# Identification of T Cell Receptors Targeting a Neoantigen Derived from Recurrently Mutated FGFR3

**DOI:** 10.3390/cancers15041031

**Published:** 2023-02-06

**Authors:** Tomohiro Tate, Saki Matsumoto, Kensaku Nemoto, Matthias Leisegang, Satoshi Nagayama, Kazutaka Obama, Yusuke Nakamura, Kazuma Kiyotani

**Affiliations:** 1Immunopharmacogenomics Group, Cancer Precision Medicine Center, Japanese Foundation for Cancer Research, Tokyo 135-8550, Japan; 2Department of Surgery, Graduate School of Medicine, Kyoto University, Kyoto 606-8507, Japan; 3Charité—Universitätsmedizin Berlin, Corporate Member of Freie Universität Berlin and Humboldt—Universität zu Berlin, Institute of Immunology, Lindenberger Weg 80, 13125 Berlin, Germany; 4David and Etta Jonas Center for Cellular Therapy, The University of Chicago, Chicago, IL 60637, USA; 5German Cancer Consortium (DKTK), Partner Site Berlin, and German Cancer Research Center (DKFZ), 69120 Heidelberg, Germany; 6Department of Gastroenterological Surgery, Cancer Institute Hospital, Japanese Foundation for Cancer Research, Tokyo 135-8550, Japan; 7Department of Surgery, Uji-Tokushukai Medical Center, Uji 611-0041, Kyoto, Japan; 8National Institutes of Biomedical Innovation, Health and Nutrition, Ibaraki 567-0085, Osaka, Japan

**Keywords:** cancer precision medicine, cancer vaccine, immunopharmacogenomics, shared neoantigens

## Abstract

**Simple Summary:**

Neoantigens are considered good targets for immunotherapy due to their tumor specificity. However, because neoantigens are unique in individual cancers, it is challenging to select personalized target neoantigens. In this study, we focused on "shared neoantigens", which are neoantigens derived from mutations observed commonly in a subset of cancer patients. We identified a shared neoantigen derived from FGFR3^Y373C^ through bioinformatics and in vitro screening. We identified that TCR-engineered T cells expressing TCRs for FGFR3^Y373C^ showed specific reactivity and cytotoxic activity against mutated FGFR3^Y373C^. We believe that immunotherapies targeting shared neoantigens would be a good approach for cancer treatment.

**Abstract:**

Immunotherapies, including immune checkpoint blockades, play a critically important role in cancer treatments. For immunotherapies, neoantigens, which are generated by somatic mutations in cancer cells, are thought to be good targets due to their tumor specificity. Because neoantigens are unique in individual cancers, it is challenging to develop personalized immunotherapy targeting neoantigens. In this study, we screened "shared neoantigens", which are specific types of neoantigens derived from mutations observed commonly in a subset of cancer patients. Using exome sequencing data in the Cancer Genome Atlas (TCGA), we predicted shared neoantigen peptides and performed in vitro screening of shared neoantigen-reactive CD8^+^ T cells using peripheral blood from healthy donors. We examined the functional activity of neoantigen-specific T cell receptors (TCRs) by generating TCR-engineered T cells. Among the predicted shared neoantigens from TCGA data, we found that the mutated FGFR3^Y373C^ peptide induced antigen-specific CD8^+^ T cells from the donor with *HLA-A*02:06* via an ELISPOT assay. Subsequently, we obtained FGFR3^Y373C^-specific CD8^+^ T cell clones and identified two different sets of TCRs specifically reactive to FGFR3^Y373C^. We found that the TCR-engineered T cells expressing FGFR3^Y373C^-specific TCRs recognized the mutated FGFR3^Y373C^ peptide but not the corresponding wild-type peptide. These two FGFR3^Y373C^-specific TCR-engineered T cells showed cytotoxic activity against mutated FGFR3^Y373C^-loaded cells. These results imply the possibility of strategies of immunotherapies targeting shared neoantigens, including cancer vaccines and TCR-engineered T cell therapies.

## 1. Introduction

In addition to surgery, chemotherapy, and radiotherapy, immunotherapy now plays a critical role in cancer management. Immune checkpoint inhibitors (ICIs), such as antibodies against programmed death-1 (PD-1), PD-1 ligand (PD-L1), and cytotoxic T lymphocyte-associated antigen 4 (CTLA-4), have considerably improved the prognosis of patients with various types of cancer, especially in cancers with "inflamed" tumors characterized by high immune cell infiltration, high PD-L1 expression, and high mutation burden [1,2,3]. However, clinical responses to ICIs have been limited to only 10%–40% of patients, and most patients experienced little or no clinical benefit. For patients not responding to ICIs due to low immune cell infiltration, a cancer vaccine or adoptive cell therapies are promising approaches to increase or induce active cancer-reactive T cells. Neoantigens, antigens derived from mutated proteins generated by somatic mutations in cancer cells, are thought to be good targets for immunotherapies because of their high specificity to tumor cells.

To facilitate neoantigen-targeting immunotherapies, we have established pipelines to predict neoantigens from patients’ genome sequencing data to identify neoantigen-specific T cells and T cell receptors (TCRs), and to generate neoantigen-specific TCR-engineered T cells [4,5,6,7,8]. Because neoantigens are not often shared among multiple cancer patients, the personalized selection of appropriate target neoantigens for individual patients is required. Recently, several reports demonstrated that "shared neoantigens", referring to mutated antigens commonly detected in a subset of cancer patients, were targeted by tumor-infiltrating T lymphocytes (TILs) [9,10,11]. Therefore, shared neoantigens could be broadly applicable targets for immunotherapies in different types of cancer. However, information about whether shared neoantigens can induce their specific cytotoxic T cells is still very limited. One of the first reports of shared neoantigens is that T cells reactive to KRAS G12D-derived peptide presented on human leukocyte antigen (HLA)-C*08:02 were identified in TILs from a pancreatic cancer patient [10]. They observed objective regression of lung metastases after the infusion of TILs composed of T cell clones that specifically targeted KRAS G12D [12] or TCR-engineered T cells targeting KRAS G12D on HLA-C*08:02 [13]. However, targetable patients by this KRAS G12D shard neoantigen were limited due to the low frequency of the *HLA-C*08:02* allele. Shared neoantigens derived from TP53 hotspot mutations have also been identified, including R175H (presented on HLA-A*02:01), Y202C (HLA-A*02:01), and R248W (HLA-A*68:01) in TILs from patients with epithelial cancers [14,15]. Although *HLA-A*02:01* is the most common *HLA* allele in Caucasians, the total frequency of these three *TP53* mutations is less than 5% in the pan-cancer population. Thus, more extensive screenings and identification of targetable shared neoantigens are required to apply this concept to a clinical setting.

In the present study, we attempted to comprehensively screen shared neoantigens derived from recurrent somatic mutations observed in 10,182 exome sequencing data in the Cancer Genome Atlas (TCGA) and identified a peptide derived from FGFR3 Y373C, which is frequent in bladder cancer, as a candidate of shared neoantigens. Here, we report the establishment of genetically engineered FGFR3^Y373C^-specific TCR-T cells and their cytotoxic functions.

## 2. Materials and Methods

### 2.1. Cell Lines and Antibodies

We purchased C1R (B lymphoblasts lacking endogenous human leukocyte antigen (HLA)-A and HLA-B expression), Jiyoye, and EB-3 cells from the American Type Culture Collection (Rockville, MD, USA) and cultured them in RPMI1640 supplemented with 10% FBS. We developed C1R cells stably expressing HLA-A*02:01 or HLA-A*24:02 (C1R-A0201 or C1R-A2402, respectively) in our previous study [4]. We generated C1R-A0206, C1R-A1101, C1R-A3101, and C1R-A3303 by the nucleofection of pCAGGS vectors encoding *HLA-A*02:06*, *HLA-A*11:01*, *HLA-A*31:01,* and *HLA-A*33:03* cDNAs, respectively.

We used the following anti-human antibodies for cell surface staining by flow cytometry analyses: We purchased CD3-APC (cat# 300311), CD3-APC-Cy7 (clone HIT3a, cat# 300317), CD4-PE-Cy7 (clone RPA-T4, cat# 300511), CD8-FITC (cat# 560960), and CD8-PE-Cy7 (clone HIT8a, cat# 301012) from BioLegend (San Diego, CA, USA). We also obtained an anti-mouse TCRβ chain-APC antibody (clone H57-597, cat# 109212) from BioLegend (San Diego, CA, USA). We purchased peptide-HLA tetramers labeled with PE from MBL (Tokyo, Japan). We analyzed all flow cytometry data using FlowJo software (ver.10, BD Biosciences).

### 2.2. Prediction of Potential Shared Neoantigens

We downloaded TCGA somatic mutation data (MAF files) called using mutect2, which consists of 10,182 samples across 33 cancer types from NCI’s Genomic Data Commons (GDC) [16]. From 1,482,002 nonsynonymous single-nucleotide variations (SNVs), we extracted 166 recurrent mutations detected in three or more cases and with a frequency of more than 1.5% in at least one cancer type (Table 1). We then predicted the binding affinities of 8- to 11-mer peptides with each amino-acid substitution to HLA class I molecules using NetMHCv3.4/NetMHCpnav2.8 software [7,17,18,19]. In this study, we targeted HLA class I alleles frequently observed (>5%) in the Japanese population [20,21]. We picked up the peptides with predicted binding affinities of <500 nM as shared neoantigen candidates. We further filtered possible shared neoantigens using our previously reported pipeline [7,8]. The shared neoantigen candidate peptides were synthesized by Innopep Inc. (San Diego, CA, USA). We used peptides for cytomegalovirus (CMV pp65 peptides for HLA-A*02:01 or HLA-A*24:02) purchased from MBL as positive controls.

### 2.3. Induction of Neoantigen-Reactive CD8^+^ T Cells Using Peripheral Blood Mononuclear Cells (PBMCs) from Healthy Donors

We induced shared neoantigen-reactive T cells following our previously reported protocols [4,22]. We purchased PBMCs from healthy donors from Cellular Technology Ltd. (Shaker Heights, OH, USA, cat# CTL-CP1) or collected them under the approved IRB after obtaining written informed consent (2018-GA-1021, 2020-GA-1090). We separated CD8^+^ T and CD8^−^ cells using the Dynabeads CD8 Positive Isolation Kit (Thermo Fisher Scientific, Carlsbad, CA, USA, cat# DB11333). We conducted the first screening using our rapid induction protocol with minor modifications [4]. Briefly, we generated monocyte-derived dendritic cells (DCs) from CD8^−^ cells using a plastic adherence method, and we cultured them in RPMI1640/AIM-V (Thermo Fisher Scientific, cat# 12055-091) containing 1% human AB serum (ABS), 500 U/mL of interleukin (IL)-4 (R&D Systems, Minneapolis, MN, USA, cat# 204-IL), and 1000 U/mL of a granulocyte–macrophage colony-stimulating factor (GM-CSF) (PeproTech, Rocky Hill, NJ, USA, cat# 300-03) for 72 h. Then we maturated DCs by adding 100 U/mL of interferon (IFN)-γ (PeproTech, cat# 300-02) and 10 ng/mL of lipopolysaccharide (LPS) (Sigma-Aldrich, ST. Louis, MO, USA, cat# L4516) and pulsed with 20 μg/mL of each of the respective neoantigen peptides for 16 h. We co-cultured and maintained the peptide-pulsed mature DCs (1.2 × 10^5^ cells) and autologous CD8^+^ T cells (5 × 10^5^ cells) in RPMI1640/AIM-V supplemented with 5% ABS and 30 ng/mL of IL-21 (R&D Systems, cat# 8879-IL) for 12 days; then, we added 5 ng/mL of IL-7 (R&D Systems, cat# 207-IL) and 5 ng/mL of IL-15 (R&D Systems, cat# 247-ILB) to the culture media every 2–3 days. We assessed the induction of neoantigen-reactive T cells using an IFN-γ Enzyme-Linked ImmunoSpot (ELISPOT) assay.

To enrich neoantigen-specific T cell clones, we performed a large-scale induction of antigen-reactive T cells based on our previous protocol [22]. Briefly, we maturated monocyte-derived DCs with 0.1 KE/mL of OK-432 (Chugai Pharmaceutical Co., Tokyo, Japan, cat# 2223496) and then pulsed them with 20 µg/mL of each peptide. We co-cultured autologous CD8^+^ T cells with mature DCs in RPMI1640/AIM-V supplemented with 5% ABS and 24 IU/mL of IL-2 for 7 days. We re-stimulated CD8^+^ T cells with the peptide-pulsed mature DCs twice (on days 8 and 15) to enrich antigen-reactive CD8^+^ T cells. We evaluated the antigen-reactivity of the expanded CD8^+^ T cells using an ELISPOT or peptide–HLA tetramer assay.

We performed limiting dilution to establish clonal T cells by seeding the CD8^+^ T cells in ~1 cell/well conditions in a round-bottom 96-well plate and by co-culturing with mitomycin C-treated Jiyoye and EB-3 cells used as feeder cells for 2 weeks. We assessed the antigen-specific reactivity of each clone by an ELISPOT assay.

### 2.4. ELISPOT Assay and Enzyme-Linked Immunosorbent Assay (ELISA)

We performed an ELISPOT assay using Human IFN-γ ELISpotPRO kit (MABTECH, 3420-2HST) or Human IFN-γ ELISPOT set (BD Biosciences, cat# 551849). Briefly, we pulsed C1R cells expressing a single HLA-A allele with each respective peptide for 16–24 h at 37 °C under 5% CO_2_. We pre-treated T cells with 30 IU/mL of IL-2 for 16 h and then co-cultured with the peptide-pulsed C1R cells (2 × 10^4^ cells/well) at 37 °C for 24 h in a 96-well plate. We captured and analyzed spots with an automated ELISPOT reader, ImmunoSPOT S6 (Cellular Technology Ltd., OH, USA). We used anti-CD3 antibody (clone CD3-2) or 50 ng/mL of phorbol myristate acetate (PMA, cat# 162-23591)/1 μg/mL ionomycin (Sigma-Aldrich, cat# 10634) as positive control wells. To measure tumor necrosis factor (TNF)-α secretion, we used an IFN-γ/TNF-α Human T cell ELISpot kit (Cellular Technology Ltd.).

We performed ELISA to measure the secreted IFN-γ levels in the supernatant using the OptEIA Human IFN-γ ELISA set (BD Biosciences). Similar to the ELISPOT assay, we co-cultured T cells with the peptide-pulsed C1R-A0206 cells in 96-well plates, and then measured cytokine concentration in the supernatants according to the manufacturer’s instructions.

### 2.5. TCR Sequencing Analysis

We performed TCR sequencing using the previously described methods [19,23]. In brief, we extracted total RNA from the ELISPOT-positive or peptide-HLA-tetramer-positive cells. We synthesized cDNAs with a common 5′-RACE adapter from the total RNA using a SMART library construction kit (Clontech, Mountain View, CA, USA). We amplified the TCRα and TCRβ cDNAs by PCR using a forward primer for the SMART adapter and reverse primers corresponding to the constant region of each of the TCRα and TCRβ sequences. After adding the Illumina index sequences with a barcode using the Nextera XT Index Kit (Illumina, San Diego, CA, USA), we sequenced the prepared libraries by 300-bp paired-end reads on the Illumina MiSeq platform using a MiSeq Reagent v3 600-cycles kit (Illumina, San Diego, CA, USA). We analyzed the obtained sequence reads using Tcrip software [23].

### 2.6. TCR-Engineered T Cells

We synthesized codon-optimized TCRα and TCRβ sequences by GeneArt (Thermo Fisher Scientific) and cloned them into the *Bam*HI and *Not*I restriction enzyme sites of the pMXs retroviral vector (Cell Biolab, San Diego, CA, USA, cat# RTV-010). To increase TCR surface expression and reduce mispairing with endogenous TCRs, we used TCRs with a mouse constant region [24]. We generated transient retroviral supernatants and transduced PBMCs from healthy donors, as previously described [25]. We examined the expression of engineered TCRs with an anti-mouse TCRβ antibody.

### 2.7. Cytotoxicity Assay

We performed the cytotoxicity assay using a CytoTox 96 Non-Radioactive Cytotoxicity Assay kit (Promega, Madison, WI, USA, cat# G1780) according to the manufacturer’s instructions. Briefly, we used C1R-A0206 cells pulsed with the respective peptides (1 μmol/L) at 37 °C for 20 h as target cells. We mixed neoantigen-specific TCR-engineered T cells (effector cells) and target cells in a 96-well plate at 1:1, 5:1, 10:1, or 20:1 ratios and incubated for 4 h at 37 °C under 5% CO_2_ condition. We conducted experiments in quadruplicate. We measured the maximum lactate dehydrogenase (LDH) release from target cells by adding a lysis solution. We measured the spontaneous LDH release of effector and target cells by separate incubation of the respective population. After 4 h incubation, we centrifuged the plate at 250× *g* for 4 min. We transferred the supernatant to a new 96-well plate. Then, we added substrate to each well before incubating the plate for 30 min in the dark at room temperature. We added a stop solution to terminate the reaction and recorded the absorbance at 490 nm. We calculated the percentage of cytotoxic activity according to the following formula:% Cytotoxicity = [(Experimental − Effector Spontaneous − Target Spontaneous)/(Target Maximum − Target Spontaneous)] × 100. 

### 2.8. Statistical Analysis

We applied the Student *t*-test to compare INF-γ secretion in ELISA and the percentage of cytotoxic activity between C1R-A0206 pulsed with mutant or corresponding wild-type peptides. We performed statistical analyses using GraphPad Prism version 9.0 (GraphPad software). We considered a *p*-value of <0.05 as statistically significant.

## 3. Results

### 3.1. Screening of Shared Neoantigens from TCGA Exome Sequencing Data

To screen neoantigens shared among multiple cancer cases, we used a total of 1,482,002 nonsynonymous SNVs detected in 10,182 exome sequencing data across 33 cancer types in the TCGA database (Table 1 and Figure 1A). We examined the numbers and frequencies of recurrent mutations in the database using different thresholds (from 0.5% to 10.0% in at least one cancer type) and selected 1.5% as the threshold most efficiently covering a broad range of patients (Appendix A and Appendix A). We picked up 166 recurrent mutations in 100 different genes, covering 39.1% of cancer cases in the TCGA database. The genes in which recurrent mutations were frequently observed were *TP53* (8.7% in all cancer types), *PIK3CA* (7.9%), *KRAS* (6.6%), *BRAF* (5.8%), *IDH1* (4.5%), *NRAS* (2.1%), *FBXW7* (1.2%), *PTEN* (0.77%), *CTNNB1* (0.77%), *EGFR* (0.64%), *GTF2I* (0.63%), *AKT1* (0.52%), *ERBB2* (0.51%), and *FGFR3* (0.45%). We then selected 1229 predicted shared neoantigen peptides to be possibly presented on at least one of the 25 HLA-A, B, or C molecules commonly observed in the Japanese population (Appendix A). Among them, 384 peptides were predicted to bind strongly with IC_50_ of less than 50 nM.

### 3.2. Induction of Shared Neoantigen-Reactive CD8^+^ T Cells Using HLA-Matched Healthy Donors’ Blood

Among the 1229 shared neoantigen candidates, we narrowed down candidate neoantigen peptides based on mutation frequency, gene function, and the frequency of HLA alleles, and finally selected 11 shared neoantigens (from the top 5 frequently mutated oncogenes), which are likely to bind to HLA-A*02:06. We examined the induction of antigen-reactive CD8^+^ T cells for a maximum of 11 peptides/sample using peripheral blood samples from five healthy donors with *HLA-A*02:06* (Figure 1B and Appendix A). In healthy donor 1, we observed a considerable peptide-dependent increase in IFN-γ spots for mutated FGFR3^Y373C^ peptide presented on C1R-A0206 cells (Figure 1B). We confirmed the FGFR3^Y373C^-reactive CD8^+^ T cells were not reactive to C1R-A0206 cells pulsed with corresponding wild-type FGFR3^WT^ peptide (Figure 1C). We found no further shared neoantigen candidates in IFN-γ ELISPOT screening using the other healthy donors (Appendix A).

### 3.3. Isolation of FGFR3^Y373C^-specific CD8^+^ T Cell Clones and Identification of Their TCR Sequences

To obtain FGFR3^Y373C^-specific CD8^+^ T cell clones, we conducted a large-scale induction of CD8^+^ T cells by co-culturing with monocyte-derived autologous DCs pulsed with FGFR3^Y373C^ for 3 weeks (Figure 2A). In all four wells (wells A to D) in the ELISPOT assay, higher IFN-γ production was observed when these CD8^+^ T cells were re-stimulated overnight with C1R-A0206 cells loaded with the mutant FGFR3^Y373C^ peptide compared with those pulsed without peptide or those with FGFR3^WT^ peptide. We further expanded these CD8^+^ T cells by co-culturing with feeder cells for two weeks and confirmed that all these CD8^+^ T cell lines secreted IFN-γ in mutant peptide FGFR3^Y373C^-specific and responder/stimulator ratio-dependent manners by ELISA (Figure 2B). We then examined the reactivities of different CD8^+^ T cell lines to the FGFR3^Y373C^–HLA tetramer by flow cytometry (Figure 2C). We observed reactivity against FGFR3^Y373C^ peptide–HLA tetramer in CD8^+^ T cells in wells C and D. Because CD8^+^ T cells in well D showed stronger reactivity, the CD8^+^tetramer^+^ T cell population (0.30%) was sorted and subjected to TCR sequencing analysis. From the results of the CD8^+^tetramer^+^ T cell population’s TCR sequencing, we found one dominant TCRβ and two dominant TCRα (Figure 2D). We could not conclude whether there were two different T cell clones with the same TCRβ or a single T cell clone expressing two different TCRα alleles [26]. Then we generated TCR-engineered T cells for both TCRα and TCRβ pairs (named TCR1 and TCR2) for further functional analyses. Because the FGFR3^Y373C^ peptide-HLA tetramer could not recognize FGFR3^Y373C^-reactive CD8^+^ T cells A and B (the discrepancy between an ELISPOT assay and an HLA multimer assay is sometimes observed), we conducted a limiting dilution by seeding FGFR3^Y373C^-reactive CD8^+^ T cells to a ~1 cell/well in two 96-well plates and a co-culture with feeder cells for two weeks. CD8^+^ T cells in three wells (wells LD1, 2, and 3) showed specific reactivity to FGFR3^Y373C^ but not to FGFR3^WT^ (Figure 2E). The TCR sequencing of these three CD8^+^ T cell lines identified two possible TCRα and TCRβ pairs; however, only one of which (named TCR3) was commonly observed in the bulk FGFR3^Y373C^-stimulated CD8^+^ T cells (Figure 2F). Hence, we used TCR3 for further functional analyses.

### 3.4. Mutant Peptide-Specific Recognition and Cytotoxicity of FGFR3^Y373C^-specific TCR-engineered T Cells

We generated TCR-engineered T cells for the three FGFR3^Y373C^ TCRs by retroviral transduction of TCRs into allogenic healthy donors’ PBMCs (Figure 3A and Appendix A). TCR1 and TCR3 specifically reacted against FGFR3^Y373C^ but not wild-type FGFR3^WT^, whereas TCR2 responded to neither FGFR3^Y373C^ nor FGFR3^WT^ (Figure 3B). We then evaluated the specificity and avidity of TCR1 and TCR3 to C1R-A0206 cells pulsed with a serial dilution of mutated FGFR3^Y373C^ and wild-type FGFR3^WT^ peptides. The ELISPOT assay showed that TCR-engineered T cells expressing TCR1 and TCR3 secreted IFN-γ and TNF-α in mutant peptide-specific and peptide-dose-dependent manners (Figure 3C,F). In the experiments using C1R cells stably expressing different HLA-A molecules, including HLA-A0201, A1101, A2402, A3101, and A3303, we confirmed IFN-γ production was almost exclusively observed when co-cultured with C1R-A0206 cells, indicating that these TCR recognitions are restricted to HLA-A*02:06 (Figure 3D,G).

To further validate the function of these TCR-engineered T cells, we performed a cell-mediated cytotoxic assay by measuring LDH release from damaged target cells. We explored the cytotoxic activity of these T cells against C1R-A0206 cells loaded with either mutant or corresponding wild-type peptides. FGFR3^Y373C^-specific TCR-engineered T cells expressing TCR1 or TCR3 showed high cytotoxic activity against C1R-A0206 cells pulsed with mutant FGFR3^Y373C^ peptide (Figure 3E,H).

## 4. Discussion

Several types of tumor-specific antigens have been investigated as targets for cancer immunotherapies [27]. Because differentiation antigens (such as MART-1 and gp100) and overexpressed antigens (such as ERBB2) were not specific to cancer cells, cancer–testis antigens expressed in cancer cells as well as in testis, such as MAGE and NY-ESO-1, have been well-studied as targets of cancer vaccines and adoptive T cell therapies [28]. Neoantigens are thought to be more cancer-specific than these antigens; therefore, they are believed to be better targets for cancer immunotherapies. Although several clinical trials of cancer vaccines targeting personalized neoantigens have been reported [29,30,31,32,33], these personalized approaches are still challenging to apply in clinical settings. In this study, we focused on "shared neoantigens" commonly targetable for a subset of cancer patients with a common HLA allele. Through comprehensive bioinformatics screening using 10,182 TCGA exome sequencing data across 33 cancer types, we picked up 1229 shared neoantigen candidates. We identified CD8^+^ T cells and their TCRs specific to the FGFR3^Y373C^-mutated peptide but not to the corresponding wild-type peptide. By generating FGFR3^Y373C^-specific TCR-engineered T cells, we found that the TCR-engineered T cells could specifically recognize mutant FGFR3^Y373C^.

FGFR3 is activated by binding to FGFs and activates downstream MAPK and PI3k/Akt/mTOR signaling and plays diverse roles in the control of cell proliferation, differentiation, and angiogenesis [34]. FGFR3 is frequently over-expressed in several cancer types and is considered an oncogene. FGFR3 somatic mutations are commonly observed in bladder cancer; furthermore, S249C is the most frequently observed mutation, and Y373C is the second most common mutation, with frequencies of 7.3% and 1.9%, respectively, in TCGA-BLCA (Appendix A). The FGFR3 Y373C mutation, located in the third immunoglobulin loop, which is important for ligand binding, is reported as an oncogenic mutation [34]. The incidence and mortality of bladder cancer are estimated at approximately 550,000 and 200,000 persons per year worldwide, respectively [35]. Hence, immunotherapy targeting the FGFR3 Y373C mutation would be one of the potential approaches for a subset of bladder cancer patients. In addition to FGFR3, PIK3CA E545K (7.0%), E542K (4.4%), and H1047R (1.5%), ERBB2 S310F (4.6%), RXRA S427F (2.9%), and several TP53 recurrent mutations are frequently found in bladder cancers (Appendix A). Neoantigen peptides derived from these mutated genes were predicted to bind at least one HLA class I molecule (range from 1 to 10) in our neoantigen prediction pipeline (Appendix A). Although more extensive screenings that target these shared neoantigens are required, this kind of approach would be useful for the future development of immunotherapy for bladder cancer patients.

The same classes of HLA have similar antigen recognition [36]. Both of the TCR-engineered T cells expressing TCR1 and TCR3 identified as FGFR3^Y373C^-specific TCRs were reactive to FGFR3^Y373C^-pulsed C1R-A0206 but not to C1R-A0201 (Figure 3D,G). However, HLA-A*02:01 and HLA-A*02:06 have only one amino acid difference, phenylalanine, and tyrosine at position 9, in the peptide binding groove [36], and our prediction system suggested the FGFR3^Y373C^ peptide might bind to HLA-A*02:01 (binding affinity IC_50_ of 291 nM; Appendix A). Several previous reports demonstrated the similarities and differences of T cell reactivity against peptides on the different HLA-A*02 molecules, including HLA-A*02:01 and HLA-A*02:06. In our previous report, we induced eight SARS-CoV-2-derived peptide-reactive CD8^+^ T cell clones from healthy donors carrying *HLA-A*02:01* [37]. Among the CD8^+^ T cell clones, seven were reactive to peptide-loaded HEV0011 cells and *HLA-A*02:06* homozygous immortalized B lymphocytes; however, the remaining one was not reactive to peptide-loaded HEV0011 cells. Similar differences were reported in hepatitis B virus (HBV) peptide-specific or MAGE-3-peptide-specific CD8^+^ T cells [38,39]. Van Buuren et al. reported that recognitions of 10 different peptide-specific T cell clones by peptide–HLA multimers were highly variable among the HLA-A*02 molecules and were not predicted by their sequence homology [40]. Moreover, GnTV_VLP_ peptide-induced specific CD8^+^ T cells from both *HLA-A*02:01*- and *HLA-A*02:06*-positive patients; however, each of the GnTV_VLP_-specific CD8^+^ T cell clones were restricted to HLA-A*02:01 and HLA-A*02:06, respectively. In the current study, although we have not confirmed the presentation of FGFR3^Y373C^ on other HLA-A02 molecules, FGFR3^Y373C^ might be able to induce specific CD8^+^ T cells in individuals carrying other *HLA-A*02* alleles, suggesting the possibility that FGFR3^Y373C^ is available as the target shared neoantigen for a wider range of bladder cancer patients. To apply the data in the current study to a clinical setting, we need further experiments. Although we have successfully identified several neoantigens using our neoantigen prediction pipeline and rapid screening system, and have confirmed that TCR-engineered T cells had the potential to eradicate tumors expressing target neoantigens [41,42], we need further confirmations that FGFR3^Y373C^-specific TCR-engineered T cells recognize and eradicate tumors expressing FGFR3^Y373C^ in animal models.

## 5. Conclusions

We identified a shared neoantigen derived from FGFR3 through bioinformatics and in vitro screening. Although more extensive screenings and further accumulation of shared neoantigen information are required, we believe that immunotherapies targeting shared neoantigens would constitute good approaches for cancer treatment.

## Figures and Tables

**Figure 1 cancers-15-01031-f001:**
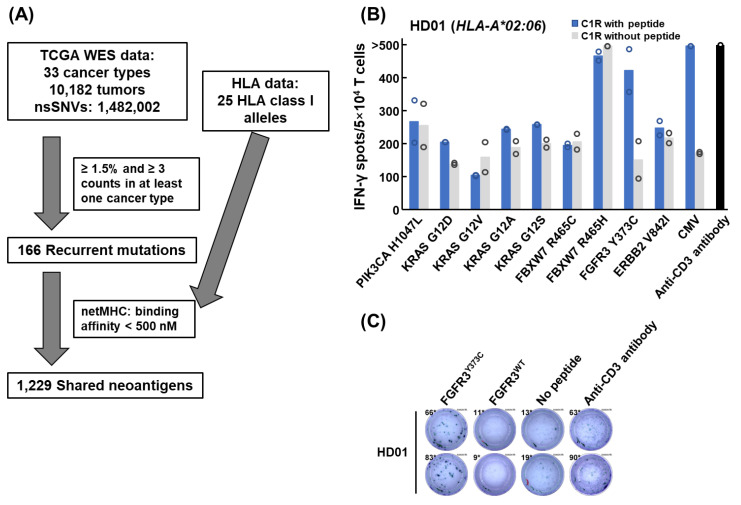
Shared neoantigen screening from TCGA exome sequencing data. (**A**) Workflow for shared neoantigen screening. From 1,482,002 nonsynonymous single-nucleotide variants (nsSNVs) in 33 cancer types in TCGA exome data, 166 recurrent mutations were detected in more than 3 tumors, and 1.5% of tumors in at least one cancer type were extracted. Then, 1229 shared neoantigen peptides were predicted to be presented on at least one of 25 HLA class I molecules, which are frequently observed in Japanese populations. (**B**) In-vitro screening of shared neoantigen-reactive CD8^+^ T cells. IFN-γ ELISPOT screening of antigen-reactive CD8^+^ T cells for 9 shared neoantigen peptides using peripheral blood of healthy donor 1 (HD01) with *HLA-A*02:06*. Antigen-stimulated CD8^+^ T cells were co-cultured with C1R-A0206 cells pulsed with or without shared neoantigen peptides. Bars represented means in duplicate experiments (each circle indicates the number of spots). (**C**) Mutant FGFR3^Y373C^ peptide-specific IFN-γ production of FGFR3^Y373C^-reactive CD8^+^ T cells. FGFR3^Y373C^-stimulated CD8^+^ T cells were co-cultured with C1R-A0206 cells pulsed with or without FGFR3^Y373C^ or corresponding wild-type FGFR3^WT^ peptide.

**Figure 2 cancers-15-01031-f002:**
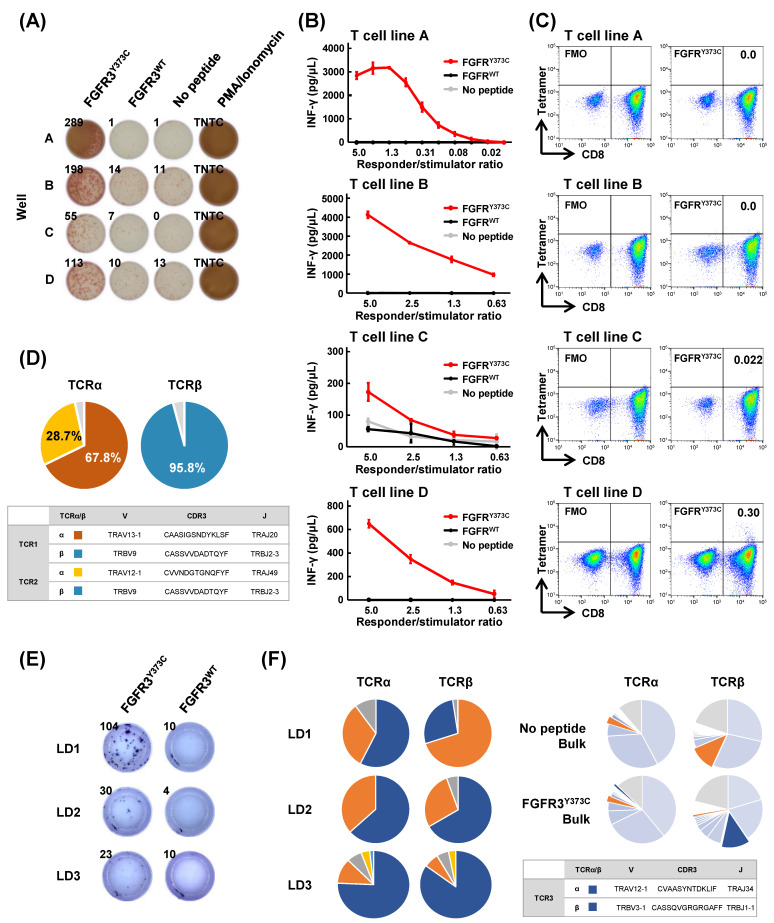
Identification of TCRs of FGFR3^Y373C^-specific CD8^+^ T cells in a large-scale screening. (**A**) A large-scale screening of mutant FGFR3^Y373C^ peptide-specific CD8^+^ T cells by IFN-γ ELISPOT assay. FGFR3^Y373C^ antigen-stimulated CD8^+^ T cells were co-cultured with C1R-A0206 cells pulsed with FGFR3^Y373C^ or wild-type FGFR3^WT^ peptide. (**B**) IFN-γ secretion of expanded FGFR3^Y373C^-specific CD8^+^ T cell lines. FGFR3^Y373C^-reactive CD8^+^ T cells in wells A to D were further expanded by co-culturing with feeder cells to generate T cell lines A to D. FGFR3^Y373C^-reactive CD8^+^ T cells (responders) were co-cultured overnight with C1R-A0206 cells (stimulators) loaded with FGFR3^Y373C^ or FGFR3^WT^ at different ratios of responders and stimulators. IFN-γ secretion from T cell lines A to D (from top to bottom) was detected by ELISA. (**C**) Representative flow cytometry plots of fluorescence minus one control (FMO; left) and FGFR3^Y373C^ peptide–HLA-tetramer (right) staining of the expanded FGFR3^Y373C^-specific T cell lines A to D (from top to bottom). CD8^+^tetramer^+^ cells were detected in T cell lines C and D. CD8^+^tetramer^+^ population detected in T cell clone D was sorted for TCR sequencing analysis. (**D**) Frequency distribution of TCRα and TCRβ sequences of CD8^+^tetramer^+^ population of FGFR3^Y373C^-specific T cell clone D. Each pie chart represents the frequency of unique TCRα and TCRβ sequences. A table summarizes TCRα and TCRβ sequences of FGFR3^Y373C^-specific TCR1 and TCR2. (**E**) Mutant FGFR3^Y373C^ peptide-specific IFN-γ production of FGFR3^Y373C^-reactive CD8^+^ T cells after a limiting dilution. (**F**) Frequency distribution of TCRα and TCRβ sequences of FGFR3^Y373C^-specific T cell clones and bulk CD8^+^ T cells after stimulated with C1R-A0206 with or without FGFR3^Y373C^ peptide. Each pie chart represents the frequency of unique TCRα and TCRβ sequences. Blue and orange pies represent possible pairs of TCRα and TCRβ. A table summarizes TCRα and TCRβ sequences of FGFR3^Y373C^-specific TCR3.

**Figure 3 cancers-15-01031-f003:**
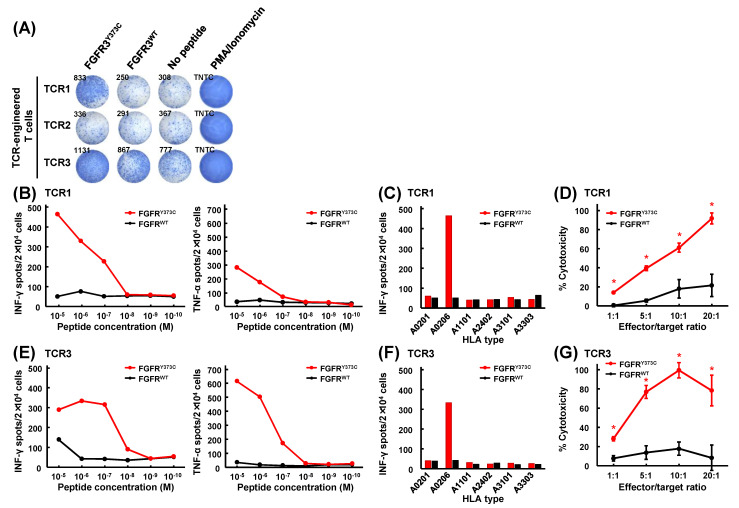
Functional assays of FGFR3^Y373C^-specific TCR-engineered T cells. (**A**) IFN-γ ELISPOT assay on FGFR3^Y373C^-specific TCR-engineered T cells generated by retroviral transduction of healthy donor’s PBMCs with TCRs. (**B,E**) Secretion of cytokines, IFN-γ (left) and TNF-α (right), of FGFR3^Y373C^ TCR-engineered T cells of TCR1 (**B**) and TCR3 (**E**) stimulated by C1R-A0206 cells loaded with graded amounts (10^−5^ to 10^−10^ M) of FGFR3^Y373C^ or wild-type FGFR3^WT^ peptides. Data are represented as means in duplication experiments. (**C**,**F**) HLA-restricted responses of FGFR3^Y373C^-specific TCR-engineered T cells expressing TCR1 (**C**) and TCR3 (**F**). IFN-γ ELISPOT assay of FGFR3^Y373C^-specific TCR-engineered T cells stimulated by co-culturing with C1R-A0201, C1R-A0206, C1R-A1101, C1R-A2402, C1R-A3101, and C1R-A3303 cells loaded with FGFR3^Y373C^ or FGFR3^WT^ peptide. (**D**,**G**) Cytotoxic activity of FGFR3^Y373C^-specific TCR-engineered T cells of TCR1 (D) and TCR3 (G) against C1R-A0206 cells pulsed with FGFR3^Y373C^ or FGFR3^WT^. The cytotoxic activity was measured in different ratios (1:1, 5:1, 10:1, and 20:1) of FGFR3^Y373C^-specific TCR-engineered T cells (effector cells) and peptide-loaded C1R-A0206 cells (target cells). Data are represented as means with standard deviations in quadruplicate experiments. Asterisks indicate statistically significant differences (*p* < 0.05) between the two groups.

**Table 1 cancers-15-01031-t001:** Summary of the number of recurrent SNVs and predicted shared neoantigen peptides in 33 TCGA cancer types.

Cancer Types	Sample Size	SNVs	Recurrent SNVs	Shared Neoantigen Peptides
ACC (Adrenocortical carcinoma)	92	5575	1	7
BLCA (Bladder urothelial carcinoma)	412	74,185	13	79
BRCA (Breast invasive carcinoma)	986	61,609	5	27
CESC (Cervical squamous cell carcinoma)	289	46,622	7	40
CHOL (Cholangiocarcinoma)	51	2174	9	49
COAD (Colon adenocarcinoma)	399	127,752	22	138
DLBC (Diffuse large B-cell lymphoma)	37	3354	3	9
ESCA (Esophageal carcinoma)	184	20,159	19	160
GBM (Glioblastoma multiforme)	393	47,827	6	35
HNSC (Head and neck squamous cell carcinoma)	508	56,481	5	19
KICH (Kidney chromophobe)	66	1546	2	6
KIRC (Kidney renal clear cell carcinoma)	336	12,977	0	0
KIRP (Kidney renal papillary cell carcinoma)	281	11,707	0	0
LAML (Acute myeloid leukemia)	143	5417	9	70
LGG (Brain lower-grade glioma)	508	19,586	10	54
LIHC (Liver hepatocellular carcinoma)	364	27,874	3	18
LUAD (Lung adenocarcinoma)	567	120,229	6	48
LUSC (Lung squamous cell carcinoma)	492	103,323	5	38
MESO (Mesothelioma)	82	1905	0	0
OV (Ovarian serous cystadenocarcinoma)	436	39,436	6	29
PAAD (Pancreatic adenocarcinoma)	178	17,272	13	80
PCPG (Pheochromocytoma and paraganglioma)	179	1335	2	14
PRAD (Prostate adenocarcinoma)	495	16,337	1	6
READ (Rectum adenocarcinoma)	137	35,319	29	207
SARC (Sarcoma)	237	12,742	1	4
SKCM (Skin cutaneous melanoma)	467	190,460	40	296
STAD (Stomach adenocarcinoma)	437	107,714	10	75
TGCT (Testicular germ cell tumors)	144	1645	4	33
THCA (Thyroid carcinoma)	492	4087	4	17
THYM (Thymoma)	123	1859	3	19
UCEC (Uterine corpus endometrial carcinoma)	530	391,093	34	254
UCS (Uterine carcinosarcoma)	57	5970	28	203
UVM (Uveal melanoma)	80	1024	6	60

## Data Availability

Available in Appendix A files.

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
