# Peer review of "Identification of T Cell Receptors Targeting a Neoantigen Derived from Recurrently Mutated FGFR3"

_cancers, 2023, doi:10.3390/cancers15041031_

Round 1

Reviewer 1 Report

Neoantigens are proteins that are derived from mutations in tumor DNA. Most of the neoantigens are unique to certain mutations in certain cancer types, which limited their potential function in immunotherapy. In this manuscript, the authors analyzed the sequencing data from the TCGA database and predicted the shared neoantigens. In addition, the authors tested the predicted peptide in vitro. Overall, the study was interesting, there are some issues that need to be addressed.

1.     Line 214, the authors mentioned that the data was from 33 cancer types. The authors should list the cancer types in a table. This information would be potentially useful for readers who are interested in certain cancer types.

2.     Lines 216-217, it was not clear how was the 1.5% threshold determined. From figure S1, it was hard to tell how 1.5% was better than other thresholds.

3.      Figure 1B, was this result significant? The authors should add the significance levels and error bars to the bar graph.  What did the circles stand for in the graph?

4.     Figure 2B, why did T cell line A and T cell line B only include the data from FGFRY373G but not the WT and No peptide controls?

Author Response

Reply to the comments from Reviewer: 1

Comments to the Author

Neoantigens are proteins that are derived from mutations in tumor DNA. Most of the neoantigens are unique to certain mutations in certain cancer types, which limited their potential function in immunotherapy. In this manuscript, the authors analyzed the sequencing data from the TCGA database and predicted the shared neoantigens. In addition, the authors tested the predicted peptide in vitro. Overall, the study was interesting, there are some issues that need to be addressed.

Comment #1:

Line 214, the authors mentioned that the data was from 33 cancer types. The authors should list the cancer types in a table. This information would be potentially useful for readers who are interested in certain cancer types.

Reply:

As suggested by the reviewer, we have listed 33 cancer types and summarized number of SNVs, recurrent SNVs and predicted neoantigens in each cancer type in Table 1 (in page 6).

Comment #2:

Lines 216-217, it was not clear how was the 1.5% threshold determined. From figure S1, it was hard to tell how 1.5% was better than other thresholds.

Reply:

We needed to select threshold to maximize patient coverage using minimum number of neoantigens to test. When we reduced the threshold to 1.0%, the values of number of covered patients/recurrent mutations, representing screening efficiency, was significantly dropped. Therefore, we selected 1.5% as threshold.

Comment #3:

Figure 1B, was this result significant? The authors should add the significance levels and error bars to the bar graph.  What did the circles stand for in the graph?

Reply:

Since we performed ELISPOT screening as “duplicate” experiments because of limited PBMCs, we have not applied statistical tests. The circles represent spot numbers in each well and bar graphs represent means of the duplicates. To avoid the confusion, we have revised description in the legend of Figure 1B as follows.

“Bars are represented as the means in duplicate experiments (each circle indicates number of spots).” (Page 7, line 270)

Comment #4:

Figure 2B, why did T cell line A and T cell line B only include the data from FGFRY373C but not the WT and No peptide controls?

 Reply:

In T cell lines A and B, IFNγ levels after stimulating with C1R pulsed with FGFRWT and no peptide were negligible (lower than 40 pg/mL) and therefore lines of FGFRWT and no peptide were overlapped with X axis.

Reviewer 2 Report

* The authors present excellent work with a novel cover for a neoantigen derived from recurrently mutated FGFR3. The manuscript is well-prepared and well-written. I have some minor issues with this manuscript.

* The background is concise and lacks some essential information such as FGFR3.

* Materials and methods section lacks catalog numbers for all used kits, chemicals, and devices.

* Results are clearly presented and the figures are well-prepared and of high quality.

* The conclusion lacks application in experimental animals with induced tumor formation.

Author Response

Reply to the comments from Reviewer: 2

Comments to the Author

The authors present excellent work with a novel cover for a neoantigen derived from recurrently mutated FGFR3. The manuscript is well-prepared and well-written. I have some minor issues with this manuscript.

Comment #1:

The background is concise and lacks some essential information such as FGFR3.

Reply:

Thank you for your suggestion. As suggested by reviewer, we have added what was previously reported and why we need to perform this study in the Introduction section. We have added several sentences with 4 references (12-15) as follows.

“One of the first reports of shared neoantigens is that T cells reactive to KRAS G12D-derived peptide presented on human leukocyte antigen (HLA)-C*08:02 were identified in TILs from a pancreatic cancer patient [10]. They observed objective regression of lung metastases after the infusion of TILs composed of T cell clones that specifically targeted KRAS G12D [12] or TCR-engineered T cells targeting KRAS G12D on HLA-C*08:02 [13]. However, targetable patients by this KRAS G12D shard neoantigen were limited due to the low frequency of HLA-C*08:02 allele. Shared neoantigens derived from TP53 hotspot mutations have been also identified, including R175H (presented on HLA-A*02:01), Y202C (HLA-A*02:01) and R248W (HLA-A*68:01) in TILs from patients with epithelial cancers [14,15]. Although HLA-A*02:01 is the most common HLA allele in Caucasians, total frequency of these three TP53 mutations are less than 5% in pan-cancer population. Thus, more extensive screenings and identification of targetable shared neoantigens are required to apply this concept to clinical setting.” (Page 2, lines 76-88)

Comment #2:

Materials and methods section lacks catalog numbers for all used kits, chemicals, and devices.

Reply:

Thank you for the comment. We have added catalog numbers.

Comment #3:

Results are clearly presented and the figures are well-prepared and of high quality.

Reply:

Thank you for your comment.

Comment #4:

The conclusion lacks application in experimental animals with induced tumor formation.

Reply:

We agreed with the reviewer’s comment to prove the efficacy of immunotherapy targeting FGFR3Y373C in vivo and therefore we have added sentences as follows.

“To apply the data in the current study to clinical setting, we need further experiments. Although we have successfully identified several neoantigens using our neoantigen prediction pipeline and rapid screening system, and have confirmed that TCR-engineered T cells had potential to eradicate tumors expressing target neoantigens [41,42], we need further confirmations that FGFR3Y373C-specific TCR-engineered T cells recognize and eradicate tumors expressing FGFR3Y373C in animal models.” (Pages 12-13, lines 439-445)

Reviewer 3 Report

Tate et al. identified FGFR3y373C as a suitable antigen for T cell therapies. Herein evidence is presented, that generation of FGFR3y373C-specific TCR-T cells is possible and that targeting FGFR3 by FGFR3y373C-specific TCR-T cells may translate into effective anticancer strategies.

The manuscript is interesting, well-structured and comprehensive and contains novel aspects for TCR-T cell therapies.

1)    FGFR3y373C-specific CD8+ T cell clones: Were different donors used for generation of FGFR3y373C-specific CD8+ T cell clones? What were the differences between the T cell clones that responded to FGFR3y373Cand those that did not?

2)    Have efforts been made to increase the transduction efficacy of TCR-engineered T cells?

3)    What was the contribution of bystander T cells to the reported cytotoxicity?

Author Response

Reply to the comments from Reviewer: 3

Comments to the Author

Tate et al. identified FGFR3Y373C as a suitable antigen for T cell therapies. Herein evidence is presented, that generation of FGFR3Y373C-specific TCR-T cells is possible and that targeting FGFR3 by FGFR3Y373C-specific TCR-T cells may translate into effective anticancer strategies.

Comment #1:

FGFR3Y373C-specific CD8+ T cell clones: Were different donors used for generation of FGFR3Y373C-specific CD8+ T cell clones? What were the differences between the T cell clones that responded to FGFR3Y373C and those that did not?

Reply:

As shown in Figure S2, we conducted an induction experiment for FGFR3Y373C in another healthy donor; however, no significant induction of FGFR3Y373C-reactive CD8+ T cells were observed. We would like to screen them using other donors in future study.

In T cell line D, single TCRβ and two different TCRα (usually one is functional and the other is not functional) expected to be expressed. We generated TCR-T cells for both TCR1 and TCR2, and confirmed TCR1 is functional TCRαβ pair but TCR2 is not functional.

Comment #2:

Have efforts been made to increase the transduction efficacy of TCR-engineered T cells?

Reply:

As pointed out by the reviewer, increasing transduction efficacy of engineered TCRs into T cells is one of the challenging steps. In our method, the transduction efficacy was usually ranging from 10% to 40%. We have confirmed reproducibility of the data in Figure 3B-3G using a different batch of TCR-engineered T cells.

Comment #3:

What was the contribution of bystander T cells to the reported cytotoxicity?

Reply:

During data processing of cytotoxicity experiments, since we subtracted the signals from non-specific cell killing, we expect the contribution of bystander T cells to the cytotoxicity in Figure 3D and 3G is limited. However, related to the comment above, we may be able to obtain more specific killing results by using TCR-engineered T cells with higher transduction efficiency.

Round 2

Reviewer 1 Report

The authors addressed my comments. The manuscript can be accepted.

Reviewer 2 Report

* The manuscript has improved greatly and are ready for publication after minor English editing revision by a native speaker.